# Mild Cystic Fibrosis Lung Disease Is Associated with Bacterial Community Stability

Thomas H. Hampton,[a] Devin Thomas,[b] Christopher van der Gast,[c] George A. O'Toole,[a] Bruce A. Stanton[a]

[a]Department of Microbiology and Immunology, Geisel School of Medicine at Dartmouth, Hanover, New Hampshire, USA
[b]Hubbard Center for Genomic Studies, University of New Hampshire, Durham, New Hampshire, USA
[c]Department of Life Sciences, Manchester Metropolitan University, Manchester, United Kingdom

**ABSTRACT** Microbial communities in the airways of persons with CF (pwCF) are variable, may include genera that are not typically associated with CF, and their composition can be difficult to correlate with long-term disease outcomes. Leveraging two large data sets characterizing sputum communities of 167 pwCF and associated metadata, we identified five bacterial community types. These communities explain 24% of the variability in lung function in this cohort, far more than single factors like Simpson diversity, which explains only 4%. Subjects with *Pseudomonas*-dominated communities tended to be older and have reduced percent predicted $FEV_1$ ($ppFEV_1$) compared to subjects with *Streptococcus*-dominated communities, consistent with previous findings. To assess the predictive power of these five communities in a longitudinal setting, we used random forests to classify 346 additional samples from 24 subjects observed 8 years on average in a range of clinical states. Subjects with mild disease were more likely to be observed at baseline, that is, not in the context of a pulmonary exacerbation, and community structure in these subjects was more self-similar over time, as measured by Bray-Curtis distance. Interestingly, we found that subjects with mild disease were more likely to remain in a mixed *Pseudomonas* community, providing some support for the climax-attack model of the CF airway. In contrast, patients with worse outcomes were more likely to show shifts among community types. Our results suggest that bacterial community instability may be a risk factor for lung function decline and indicates the need to understand factors that drive shifts in community composition.

**IMPORTANCE** While much research supports a polymicrobial view of the CF airway, one in which the community is seen as the pathogenic unit, only controlled experiments using model bacterial communities can unravel the mechanistic role played by different communities. This report uses a large data set to identify a small number of communities as a starting point in the development of tractable model systems. We describe a set of five communities that explain 24% of the variability in lung function in our data set, far more than single factors like Simpson diversity, which explained only 4%. In addition, we report that patients with severe disease experienced more shifts among community types, suggesting that bacterial community instability may be a risk factor for lung function decline. Together, these findings provide a proof of principle for selecting bacterial community model systems.

**KEYWORDS** cystic fibrosis, lung infection, microbial communities

Cystic fibrosis (CF) is a genetic disease affecting about 30,000 people in the United States and 70,000 worldwide (1). It has long been recognized that airway infection by *Pseudomonas aeruginosa* predicts morbidity and mortality in young children with CF (2). Direct sampling of the CF airway demonstrates that a small set of established CF pathogens such as *Pseudomonas aeruginosa*, *Staphylococcus aureus*, *Haemophilus influenzae*, and species from the genera *Burkholderia*, *Stenotrophomonas*, and *Achromobacter*

Address correspondence to George A. O'Toole, georgeo@dartmouth.edu, or Bruce A. Stanton, Bruce.A.Stanton@dartmouth.edu.

Bacterial communities as pathogenic units in CF lung disease

dominate early infections in children with CF (3). The prevalence of *Pseudomonas* infection in the CF community increases with age (4), and it has been reported that few pathogens other than *Pseudomonas* are detectable in end-stage disease (5). Not surprisingly, the prevailing thought was once that CF lung disease might be caused by a single pathogen (6, 7), although not necessarily the same pathogen in all people (8). Consequently, treatments were directed at specific pathogens such *P. aeruginosa*, *S. aureus*, and *H. influenzae* (9). Evidence from the last 15 years demonstrates that genera outside the established set of CF pathogens, such as *Streptococcus* and anaerobes like *Prevotella* and *Veillonella*, also play important roles in lung disease for persons with CF (pwCF) (10–12).

CF lung disease is now considered a polymicrobial phenomenon (13) that can be described using ecological concepts like species diversity. Diversity has been associated with improved lung function (14–17), but increased diversity may also signal the onset of exacerbation (18). Longitudinal analysis, in which a pwCF is followed over time, generally reveals that the CF airway microbiome is relatively stable (18–21) and changes modestly during cycles of antibiotic treatment and pulmonary exacerbations, acute phenomena characterized by a rapid decline in overall health, accompanied by decreased lung function (19, 22–24). During exacerbations, pwCF experience rapid decline in overall health, frequently accompanied by permanently decreased lung function (19). More frequent exacerbations, and failure to recover fully from these exacerbations, are thought to differentiate people with moderate or severe disease from those with mild CF lung disease (25, 26). Intravenous antibiotic treatment in a hospital setting is recommended during a pulmonary exacerbation and recovery usually occurs over a period of weeks (21). Complete recovery of lung function to levels prior to exacerbation is more common in younger pwCF and those with fewer exacerbations (22). Recovery may be mediated by response to antibiotics like tobramycin, which is at least partly predictable from polymicrobial signatures (23). It is possible that some exacerbations are triggered by increasing relative abundance of anaerobes in the two weeks leading up to an exacerbation (24) or by specific species of *Streptococcus*, including *anginosus* group streptococci (AGS), also known as the *milleri* group streptococci (25–27). Collectively, recent findings paint a complex picture of how exacerbations occur in CF.

One model for how dynamic changes in the CF airway community might be associated with changes in patient health suggests that the CF airway microbiome is normally a stable climax community (28) dominated by a classic pathogen such as *Pseudomonas*, *Achromobacter*, or *Staphylococcus*. From time to time, the intrusion of facultative or obligate anaerobes, via an undescribed mechanism, leads to a fermentative "attack community" that lowers pH and favors further expansion of anaerobes (28). This population shift would increase bacterial species diversity, consistent with the increased diversity during exacerbation that has been previously observed (18). Interestingly, this is one instance where increased diversity is associated with a worse clinical outcome rather than better outcomes, as is typically reported (14, 17, 29–31). Furthermore, the climax-attack model (CAM) is also consistent with the concept of stable core and dynamic satellite taxa (16), because these satellite taxa are largely the same genera that are thought to constitute the attack community and it has been shown that satellite genera are more dynamic across disease states (32).

The primary aim of this study was to improve our understanding of the baseline community types, including (i) identifying discrete CF airway community types; (ii) assessing whether community types are associated with different health outcomes; (iii) describing how community types in pwCF change over time; and (iv) whether community stability is associated with disease severity. This analysis included a variety of machine learning techniques applied to both cross-sectional and longitudinal data sets. Collectively, our results are consistent with a recent report that bacterial communities in stable patients have less variation in community structure over time (33), and our results support the climax-attack model of CF airway disease (28). In addition, we

identify a five-community model that accounts for 24% of the variability in lung function, far more than any single factors explored previously.

## RESULTS

**Five clusters to explain two large cross-sectional data sets.** As the emphasis in CF airway microbiome investigation has shifted toward a polymicrobial interpretation (13), data analysis has increasingly focused on multivariate comparisons. We therefore began our analysis to identify polymicrobial community clusters by calculating Bray-Curtis dissimilarity (34) between genus level taxonomic assignments in bacterial 16S rRNA gene amplicon sequencing data from 167 sputum samples from people with CF. Bray-Curtis dissimilarity is a common measure used to compare microbial communities and places a larger weight on species that are commonly observed, partly mitigating the problem of random sampling errors associated with rare species. Samples for this analysis were chosen as a cross-sectional subset of two recently published studies (17, 18) and included clinically stable subjects aged 8 to 69 from 14 CF centers in the United States and Europe, creating a snapshot that should include many of the community types present in pwCF. We calculated the Bray-Curtis dissimilarity across 167 cross-sectional samples from these two independent studies: (i) a cross-sectional study from Cuthbertson et al. (17), from which 83 samples were analyzed, and (ii) a longitudinal study by Carmody et al. (18), from which we selected a single representative sample from each patient for this analysis (84 samples).

We next used k-means to cluster samples using the Bray-Curtis dissimilarity matrix we generated above. The k-means machine learning algorithm requires the user to identify how many clusters (k) are to be identified in the data, then uses an iterative approach to assign samples to each cluster to minimize distances between samples in the same cluster and maximize distances between samples in different clusters. Such machine learning approaches are commonly applied to microbiome studies (35). For example, unsupervised techniques like hierarchical clustering (36) have been used to identify core taxa in the CF airway (16) and gut (37) and to demonstrate that samples from the same person tend to be similar to each other, even across disease and treatment (38). K-means analysis has also been used to identify groups of patients with similar patterns in inflammation, microbiota, or clinical factors (39). What makes k-means unsupervised is that similarity between clusters is based only on observed data, in this case, counts of bacterial genera. Unsupervised learning does not take sample labels (e.g., "severe disease" or "exacerbation") into consideration. Therefore, when unsupervised clustering places samples labeled "severe disease" together, it is likely that a nonrandom process (e.g., the result of a biological difference) caused the samples to cluster.

As described above, the k-means algorithm requires the user to identify how many clusters (k) are to be identified in the data. Since we had no prior information to justify a particular choice of k, we used another machine learning approach, the gap statistic (40), to estimate the number of distinct clusters in our cross-sectional data. The gap statistic has recently been used to evaluate the optimal number of clusters in CF airway microbiome data (29). The gap statistic identifies the smallest value of k that minimizes distances between samples in the same cluster and maximizes distances between samples. Gap analysis suggested the existence of five distinct community types in our cross-sectional data set of the 167 baseline samples analyzed here. We found that subjects belonging to different communities also differed significantly in lung function (see below); this finding suggests that clusters are based on biological differences (39) and that community types may be clinically relevant. The optimal number of clusters (five) chosen by the gap statistic algorithm is shown by the vertical dashed line in Fig. 1A. The basis of this choice is driven by the break in the data after $k = 5$. Specifically, before $k = 5$, the gap statistic increases, but at $k = 6$, the gap statistic decreases. It is this deflection, however brief, that the algorithm is designed to detect, because the goal is to identify the smallest k that is supported by the data. A parsimonious choice of k serves our

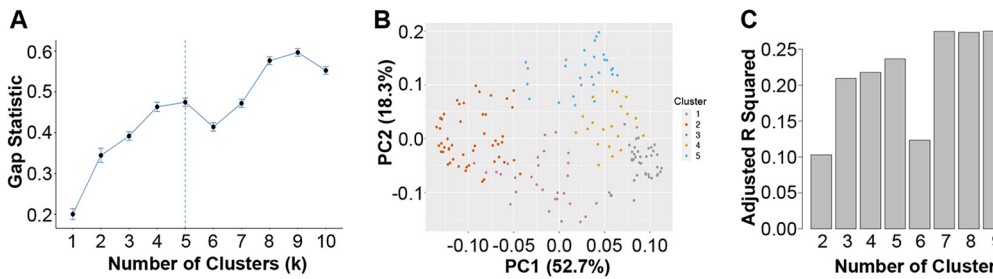

**FIG 1** Determining cluster numbers. (A) Gap statistic for different numbers of k clusters, ranging from 1 to 10. Larger values indicate greater separation between clusters. An optimal number of clusters based on gap statistic is shown by a dashed vertical line. (B) Principal coordinate analysis of 5 clusters, based on the Bray-Curtis dissimilarity matrix. The legend showing the color code of each cluster is on the right. Together, the first two principal coordinates capture >70% of the variance in these data. Each dot represents one of 167 samples in the cross-sectional analysis described in the Materials and Methods. (C) Adjusted $r$ squared from linear models of lung function (ppFEV$_1$) using membership in a given cluster as the independent variable. Models were run for different numbers of k clusters, ranging from 2 to 10.

purposes as well, as we seek to identify a small set of distinct communities with relatively few members to eventually validate in a laboratory setting.

A principal coordinate analysis that captures >70% of the variance of these data (Fig. 1B) shows how the clusters separate into 5 largely nonoverlapping groups. It is important to note that this analysis raises the possibility of alternative, potentially equally valid, grouping of the samples and/or potential overlap among these communities.

It is reassuring that choosing 5 clusters also optimizes the ability of communities to explain patient differences in lung function. Figure 1C shows the value of adjusted $r^2$ from linear models of ppFEV$_1$, a measure of lung health, as a function of cluster membership. Five clusters explain about 24% (adjusted $r^2 = 0.24$) of patient differences; using six clusters reduces $r^2$. In summary, two independent lines of evidence suggest that as few as 5 clusters may be optimal for summarizing these data.

**Cluster membership predicts lung function better than *Pseudomonas* relative abundance, age, or community diversity.** Despite a reframing of the conversation surrounding the CF airway microbiome away from single pathogens toward ecological descriptions involving species diversity (14, 17, 18, 41, 42), the relationship between *Pseudomonas* and lung function is still frequently cited (14, 23, 42, 43). We used linear models to assess the level of association between lung function and commonly identified explanatory variables such as the relative abundance of *Pseudomonas*, as well as Simpson diversity and age. Although sex and cystic fibrosis transmembrane conductance regulator (CFTR) mutation were present in the van der Gast data set (17), these annotations were not part of the Carmody data set (18), therefore these covariates were not included in our analysis. As shown in Fig. 2, single variables like relative abundance of *Pseudomonas*, Simpson diversity, and age explain less than 11% of patient variability in lung function (adjusted $r^2 < 0.11$). In other words, over 89% of variability

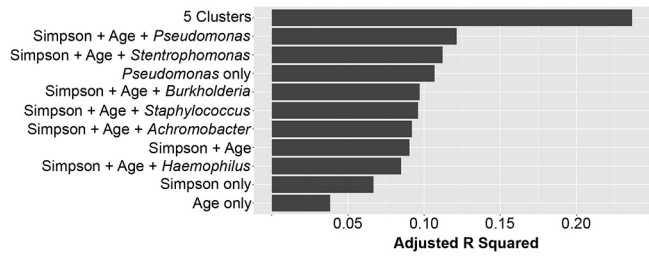

**FIG 2** Microbial clusters are the best predictor of airway function. The ability of linear models to predict ppFEV$_1$, as estimated by adjusted $R$ squared. Individual factors or combinations (denoted by "+") of covariates are shown on the y axis: Simpson diversity only, age only, *Pseudomonas* only, the combination of Simpson and age, or Simpson and age and the relative abundance of other classic CF pathogens, compared to membership in one of the 5 clusters.

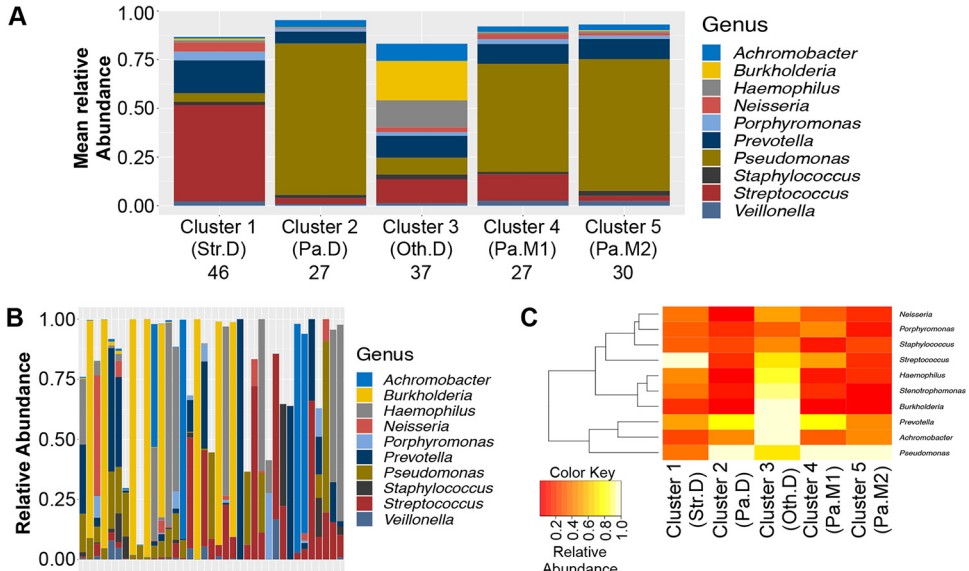

FIG 3 Analysis of five identified clusters. (A) Mean abundances of the top 10 genera in each of the 5 clusters. Mnemonic abbreviations: Str.D, *Streptococcus* dominated; Pa.D, *Pseudomonas* dominated; Oth.D, Other dominated; Pa.M1, *Pseudomonas* mixed community number 1; Pa.M2, *Pseudomonas* mixed community number 2. The number of samples that belonged to each cluster appears beneath each cluster mnemonic. (B) Abundances of the top 10 genera for samples belonging to cluster 3 (Oth.D) in each of the 37 samples belonging to this group. (C) Maximum abundance of 10 genera achieving the highest relative abundance in any of the 167 samples, as a function of cluster.

in lung function is explained by factors other than *Pseudomonas*, Simpson diversity, or age. The explanatory power of these models improves with increasing complexity, but a combined model that incorporates Simpson diversity, age, and the relative abundance of *Pseudomonas* or other classical CF pathogens in a single model still only explains less than 12% of patient variability in lung function. In contrast, membership in one of the 5 communities identified above explains 24% of patient variability in lung function, without including any other covariates. Adding age or Simpson diversity to the community model did not significantly improve its predictive power. In other words, knowing which cluster a person's airway microbiome belongs to provides a better estimate of their lung function than knowing the relative abundance of *Pseudomonas* in their airway microbiome, the diversity of their community, or their age (Fig. 2).

**Top 10 bacterial genera across clusters.** We identified the top 10 genera with the highest mean relative abundance across the 167 cross-sectional samples and averaged their relative abundance in each of the 5 clusters, as shown in Fig. 3A. These 10 genera accounted for 87% of the relative abundance in cluster 1 (Str.D), to which 46 samples belonged. This cluster was dominated by *Streptococcus* (brick red). It is noteworthy that this cluster was almost exclusively identified in the Carmody (18) data set, which contained a younger group of subjects. Cluster 2 (Pa.D, 27 samples) was dominated by *Pseudomonas* (olive), cluster 3 (Oth.D, 37 samples) was dominated by a genus other than *Pseudomonas* or *Streptococcus* and could be described as having an even mix of genera based on averages, and cluster 4 (27 samples, Pa.M1) and cluster 5 (30 samples, Pa.M2) were both dominated by *Pseudomonas* but were more mixed in nature than Pa. D. The three *Pseudomonas* clusters differ in abundance and/or type of minor genera. While cluster 4 (Pa.M1) and cluster 5 (Pa.M2) were similar, they clustered separately, perhaps because cluster 5 had more *Achromobacter* and *Burkholderia* than cluster 4 ($P < 0.001$ asymptotic Wilcoxon-Mann-Whitney test), and cluster 4 had more *Streptococcus* ($P < 0.001$, asymptotic Wilcoxon-Mann-Whitney test).

We then mapped each of these clusters back to the two cross-sectional data sets, as shown in Table 1. The Carmody data set (18) was from a single CF center in the United States, and the Cuthbertson data set (17) included subjects from Europe and the

Microbiology
Spectrum

**TABLE 1** Mapping the clusters back to the data sets

| Cluster no. | Mnemonic | Carmody | Cuthbertson |
|---|---|---|---|
| 1 | Str.D | 45 | 1 |
| 2 | Pa.D | 0 | 27 |
| 3 | Oth.D | 19 | 18 |
| 4 | Pa.M1 | 19 | 8 |
| 5 | Pa.M2 | 1 | 29 |

United States, spanning 13 different CF centers. Some clusters, but not others, were enriched in data from particular studies, perhaps as a consequence of biases in sample collection/processing, differences in subject age, and/or systematic differences in patient care. Of note, the two *P. aeruginosa* mixed communities, Pa.M1 and Pa.M2, largely clustered as a function of the data set from which they were derived. Interestingly, the Carmody (18) data set had no patients in the *P. aeruginosa* dominant cluster (cluster 2, Pa.D), while the Cuthbertson (17) samples, collected over 13 centers, had ~one-third of its samples map to this community cluster. In contrast, for the communities assigned to the Carmody data set (18), more than half were described as *Streptococcus*-dominated, while the remaining samples, though containing *P. aeruginosa*, were more evenly mixed (Pa.M1) or dominated by other pathogens (Oth.D).

It is likely that both technical and clinical differences contribute to compositional differences observed between studies. On the one hand, it is well established that CF practices and outcomes differ by region, both within and between continents (44), suggesting that CF airway microbiomes may be systematically different by region. On the other, it is well established that technical differences can cause apparent differences in community composition (45). A future study of regional CF airway microbiome differences, one in which all samples were processed identically, would be required to determine whether microbiomes truly cluster by geographic location.

Interestingly, averages do not capture the specifics of a group perfectly, as shown in Fig. 3B. Individual samples that mapped to cluster 3 (Oth.D; Other dominated), which, when averaged, appear to have an even mix of genera, tended to be dominated by individual genera, including *Stenotrophomonas*, which was not among the top 10 most abundant genera across all 167 samples. A sample-by-sample analysis of dominance revealed that *Haemophilus*, *Stenotrophomonas*, *Burkholderia*, *Prevotella*, and *Achromobacter* all achieved high levels of dominance (relative abundance greater than 85%) in cluster 3, but not in other clusters (Fig. 3C). *Streptococcus* achieved dominance in cluster 1, but not elsewhere.

**Clusters dominated by *Streptococcus* and *Pseudomonas* differ in lung function and age.** It has previously been observed that species from the *Streptococcus* genus are part of the core CF airway microbiome (16, 31, 43), and that the prevalence of *Streptococcus* spp. is sometimes associated with less severe lung function loss (12, 14, 31, 46), but that some genera of *Streptococcus*, including the *Streptococcus anginosus* group, may facilitate exacerbation (25–27). Our cross-sectional analysis highlights the positive aspect of patients carrying the community dominated by *Streptococcus*; subjects whose airway microbiomes were characterized as *Streptococcus*-dominated tended to have better lung function, as shown in Fig. 4A. Clusters differ significantly in ppFEV$_1$ after correcting for age in this model. *Streptococcus*-dominated samples tended to come from younger subjects (Fig. 4B).

**Communities observed in subjects with mild versus moderate/severe disease are similar.** Findings in the cross-sectional data (Fig. 4A and B) are consistent with previous reports showing that communities dominated by *Pseudomonas* tend to be associated with older subjects and reduced lung function. Given that young, healthier pwCF tend to have sputum communities dominated by *Streptococcus* led to the hypothesis that people with such *Streptococcus*-dominated communities ultimately transition to communities dominated by *Pseudomonas* as they age. To probe this idea, we used a longitudinal data set to assess how communities changed in the 24 subjects

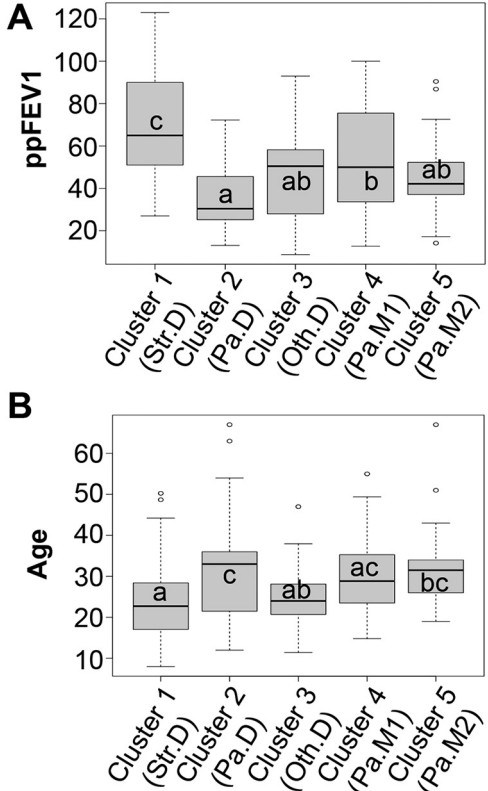

**FIG 4** Association of communities with lung function and age. (A) Association of communities with lung function. Subject-lung function profiles of 5 distinct communities. Box and whisker plot comparing ppFEV$_1$ between *Streptococcus*-dominated (Str.D), *Pseudomonas*-dominated (Pa.D), communities dominated by other genera (Oth.D), *Pseudomonas* mixed community number 1 (Pa.M1), and *Pseudomonas* mixed community number 2 (Pa.M2), as shown in Fig. 3A. Communities sharing a common letter are not significantly different; those with no letter in common differ significantly (Tukey honest significant difference, $P < .05$). For example, subjects with Str.D communities had higher ppFEV$_1$ than those with any other community type. (B) Association of communities with age. Subject-age profiles of 5 distinct communities. Box and whisker plot comparing age between *Streptococcus*-dominated (Str. D), *Pseudomonas*-dominated (Pa.D), communities dominated by other genera (Oth.D), *Pseudomonas* mixed community number 1 (Pa.M1), and *Pseudomonas* mixed community number 2 (Pa.M2), as shown in Fig. 3A. Communities sharing a common letter are not significantly different; those with no letter in common differ significantly (Tukey honest significant difference, $P < .05$). For example, subjects with Str.D communities were younger than those with Pa.D communities or Pa.M2 communities but did not differ significantly in age from subjects with Oth.D or Pa.M1 communities.

detailed in Table 2, for whom there are 10 or more longitudinal samples collected for times periods ranging from ~6 to 10 years (average of ~8 years).

For this analysis, we used supervised machine learning to track community type changes in individuals with CF through time. Briefly, we used random forests (47) to identify features that reliably classify a sample as belonging to one of the five previously identified communities. This classifier enabled us to identify the community type of 346 16S rRNA gene amplicon data sets from 24 subjects that were observed across many years (18). This longitudinal data set included 16S rRNA gene amplicon data set measurements made in different clinical states (baseline, exacerbation, treatment, and recovery) and includes clinical metadata such as whether a subject had been diagnosed as having mild or more severe disease. This approach mapped the 346 16S rRNA gene amplicon data sets to the four communities shown in Fig. 3, specifically, Str.D., Pa.D, Pa.M1, and Oth.D. No samples in the Carmody (18) longitudinal data mapped to Pa.M2, consistent with the description of the Pa.M2 cluster being specific to the Cuthbertson (17) data set (see Table 1).

Roughly two or three community types were required to describe any given subject over time, recapitulating previous reports that the CF airway microbiome is fairly stable

**TABLE 2** Characteristics of 24 subjects with at least 10 longitudinal samples[a]

| Isolate | Aggressiveness | n | Period (yr) | Median age | Median FEV1 | Clinical states[b] | FEV1/yr[c] |
|---|---|---|---|---|---|---|---|
| 75 | Moderate/Severe | 16 | 8.99 | 14.9 | 32.5 | R, E, T, T, E, R, B, B, R, T, E, B, B, R, B, R | −2.23 |
| 139 | Moderate/Severe | 15 | 6.64 | 15.1 | 34 | B, B, B, T, R, T, B, B, B, B, R, B, T, T, R | −5.17 |
| 94 | Moderate/Severe | 13 | 9.16 | 17.2 | 39 | T, T, E, E, T, B, E, B, B, B, B, R, R | 1.56 |
| 229 | Moderate/Severe | 11 | 7.27 | 19.2 | 34 | R, R, R, B, B, T, R, B, T, B, B | −4.50 |
| 52 | Moderate/Severe | 18 | 9.74 | 23.0 | 43.5 | T, E, R, R, T, R, R, B, R, R, B, E, E, B, B, B, E, T, R, R | −1.84 |
| 179 | Moderate/Severe | 24 | 8.66 | 23.8 | 41 | B, E, T, T, B, E, T, R, B, B, E, B, R, R, E, R, T, B, T, E, R, B, E, E | −3.08 |
| 159 | Mild | 11 | 7.97 | 24.3 | 82 | B, B, B, E, B, B, B, B, B, B, R | −1.20 |
| 87 | Mild | 17 | 7.54 | 25.0 | 82 | T, T, R, B, E, B, B, B, B, R, T, R, B, R, B, B | 1.84 |
| 160 | Mild | 13 | 8.01 | 25.2 | 73 | B, T, B, B, R, B, R, B, R, B, T, E, B | 0.79 |
| 142 | Moderate/Severe | 20 | 10.14 | 25.4 | 67 | B, T, T, T, T, E, B, R, T, T, E, E, R, B, T, E, B, R, T, E | −1.16 |
| 141 | Mild | 12 | 8.91 | 26.3 | 85.5 | R, E, R, B, B, B, R, T, B, R, B, T | −0.20 |
| 124 | Moderate/Severe | 18 | 10.24 | 26.3 | 54.5 | B, B, T, B, T, B, E, T, B, E, B, E, B, E, T, B, E, B, B | −1.00 |
| 147 | Moderate/Severe | 18 | 8.17 | 27.2 | 36.5 | B, R, E, E, B, B, B, B, T, R, T, T, T, B, B, R, E | −3.73 |
| 209 | Moderate/Severe | 16 | 7.87 | 28.0 | 37.5 | T, B, T, B, E, T, R, E, R, R, R, T, R, R, B, R | −6.97 |
| 53 | Mild | 10 | 5.35 | 28.6 | 69.5 | E, R, T, E, B, E, B, E, B, NA, NA | 1.55 |
| 182 | Mild | 14 | 5.69 | 29.0 | 83.5 | T, B, R, E, E, B, B, B, B, B, E, E, E | −2.28 |
| 118 | Mild | 10 | 7.33 | 30.3 | 58.5 | E, T, B, R, B, B, B, T, R, E | −1.33 |
| 244 | Mild | 10 | 6.42 | 30.8 | 74.5 | E, R, B, B, R, B, B, B, E, B | −2.23 |
| 222 | Mild | 10 | 7.47 | 33.5 | 97 | B, R, B, T, B, T, B, B, B, NA | −0.21 |
| 136 | Mild | 15 | 7.6 | 36.8 | 57 | B, R, B, E, B, B, T, R, B, R, E, E, B, T, B | −2.99 |
| 119 | Mild | 14 | 7.72 | 37.1 | 36 | E, B, T, B, B, B, T, B, B, B, R, B, B | −3.42 |
| 195 | Mild | 11 | 8.24 | 41.4 | 54 | E, B, E, E, B, E, B, T, R, B, R | 0.45 |
| 50 | Mild | 10 | 6.27 | 43.5 | 70.5 | B, E, R, B, E, B, B, B, B, R | −2.27 |
| 117 | Mild | 20 | 9.72 | 49.6 | 51 | E, E, B, R, B, E, B, E, T, T, B, T, B, E, B, E, T, T, B, B | −0.31 |

[a]Disease aggressiveness for each subject was based on publicly available metadata; n represents the number of sputum microbiomes available for each subject, observed over the period, given in years.

[b]Clinical states observed: B, baseline; E, exacerbation; T, treatment; R, recovery.

[c]Linear regression estimates of the annual change in FEV1 percent predicted for each subject shown at far right. The table is sorted by increasing median age.

over time (15, 18, 19, 39) but may shift in structure around the time of exacerbation or in response to antibiotic therapy (23, 39). We did not observe large differences in community types between subjects with mild disease compared to moderate/severe disease either as a function of subject (Fig. 5A) or disease state (Fig. 5B). For example, subject 117 was observed 15 times to show a *Streptococcus*-dominated community (Str.D) and 5 times to show the *Pseudomonas* mixed community number 1 (Pa.M1).

We did note some associations. For example, domination by a pathogen other than *Pseudomonas* (Oth.D) was associated with decreased probability of being observed in the baseline state (P value = 0.004, odds ratio 0.43, Fisher's exact test). Furthermore, despite similarities, subjects with mild disease did differ from those with moderate/severe disease in specific ways. Subjects with mild disease were more likely to be observed having a community classified as Pa.M1 (P value = 0.02, odds ratio = 1.69, Fisher's exact test) and more likely to be observed in the baseline state (P value = 3.1e-06, odds ratio = 2.8, Fisher's exact test) than people with moderate/severe disease.

Interestingly, we note that people with mild disease have bacterial communities that are more stable over time based on being observed in a specific community category (Fig. 5). We tested this association using a continuous variable by calculating Bray-Curtis dissimilarity for all pairwise comparisons of samples taken from any given subject over time. As shown in Fig. 6A, communities from subjects with mild disease were less dissimilar to each other than communities from subjects with moderate/severe disease, suggesting that variation in community composition is associated with worse disease outcomes.

As a second approach to assess the association between community stability and clinical outcome, a Markov chain analysis was performed to track subjects and their associated community types across time. Markov models simplify dynamic systems by representing them as a set of finite states and transitions between these states. With enough data, the geometry of these systems (i.e., states that are connected to each other, indicating the possibility of a transition) can be determined. In this context, Markov models can be used to identify parallels between our longitudinal cohort and accepted models of ecological succession in the cystic fibrosis airway (48).

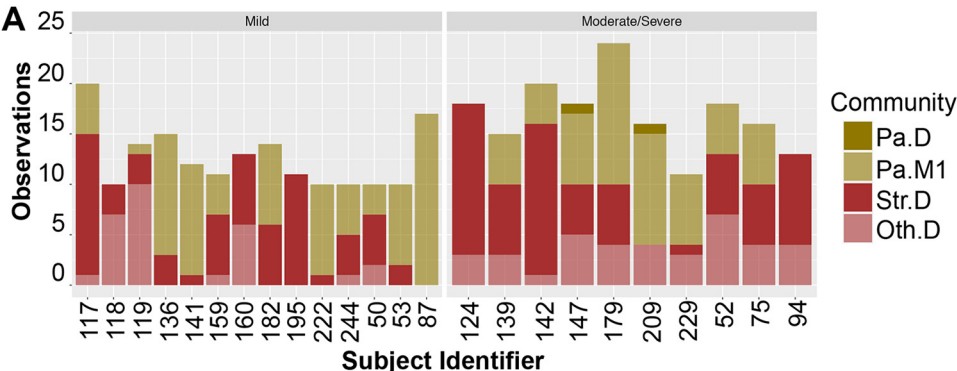

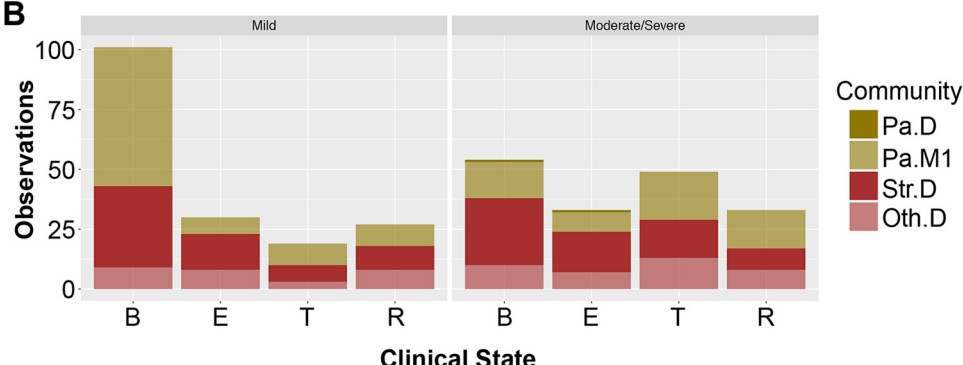

**FIG 5** Association of clusters across samples and disease state. (A) Frequency of observing various community types in subjects designated having mild or moderate/severe disease. Subjects with mild disease ($n = 14$, left) are compared to subjects with moderate/severe disease ($n = 10$, right). "Subject identifier" corresponds to "isolate" in Table 2, a field provided in the Carmody (18) data set. Community types identified in these longitudinal samples were deemed similar to those in Fig. 3A by the random forest classifier. (B) Frequency of observing various clinical states for the 14 subjects with mild disease aggressiveness (left) and the 10 subjects with moderate/severe disease (right), as shown in Table 2. Clinical states are defined as B, baseline; E, exacerbation; T, treatment; and R, recovery. Community types identified in these longitudinal samples were deemed similar to those in Fig. 3A by the random forest classifier.

Certain aspects of succession in the CF airway are well accepted; airways begin mostly sterile, are colonized by bacteria from the environment, and develop unique community structures in different pwCF. These communities are surprisingly stable in the short term but, over longer periods, lung function and diversity tend to decrease (31) as a typical CF pathogen becomes increasingly dominant (17). What is less well understood is the role played by genera that are not typically identified as CF airway pathogens, and how different communities might affect CF disease, e.g., by causing pulmonary exacerbations. The climax-attack model (49) proposes the existence of relatively benign climax communities and attack communities that cause exacerbations and remodeling. Interestingly, the model proposes that these two communities can coexist in the same subject and may contain substantially similar genera. The more virulent attack community is thought to be metabolically distinct, and to be enriched in genera that can flourish in oxygen-deprived conditions (28). Therefore, remaining in the less virulent climax community should be advantageous, even if climax community is associated with older age and reduced lung function. Our data support this assessment, although it should be noted that our communities were selected by a data-driven process and were not specifically designed to reflect climax or attack communities *per se*. In addition, the Str.D community, which would be most similar to previously described attack communities, did not significantly predict exacerbation state, as shown in Fig. 5B.

Nonetheless, the dynamic community structure detected by Markov chains is broadly consistent with cyclical behavior predicted by the climax-attack model, and the proposition

Microbiology Spectrum

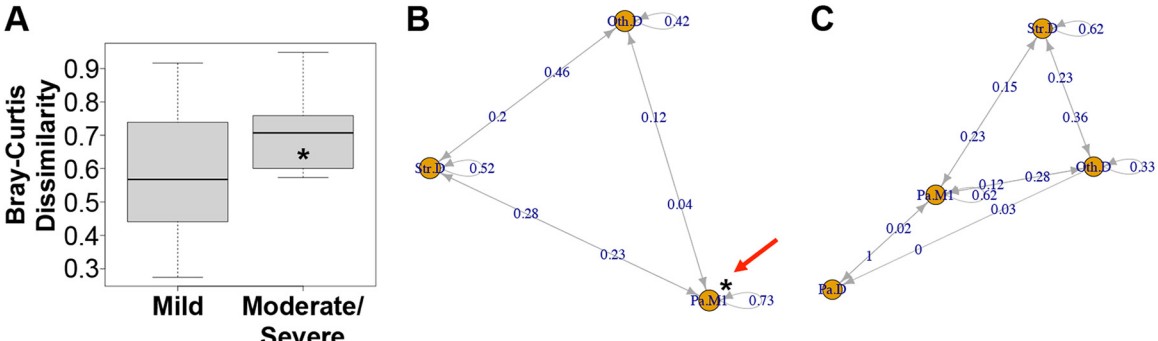

**FIG 6** Support for the climax-attack model. (A) Median within-subject pairwise distances (Bray-Curtis dissimilarity) for $n = 14$ subjects with mild disease aggressiveness (left) and $n = 10$ subjects with moderate/severe disease (right) as shown in Table 2. Subjects with mild disease had samples that were significantly more similar to each other (* signifies $P < 0.05$, Wilcoxon rank sum exact test). (B) Markov chain diagram for community transitions in subjects with mild disease (see Table 2). Each node in the diagram (orange circles) represents the state of being observed with a specific bacterial community. Edges leading from each node represent possible transitions, including the possibility of observing the same community again (loop). Numbers on arrows are estimates of the probability of taking a specific path. Subjects with mild disease were significantly more likely to remain in Pa.M1 than subjects with moderate/severe disease (red arrow; * indicates $P = 0.03$, Fisher's exact test, odds ratio 1.78). (C) Markov chain diagram for community transitions in subjects with moderate/severe disease (see Table 2). Each node in the diagram (orange circles) represents the state of being observed with a specific bacterial community. Edges leading from each node represent possible transitions, including the possibility of observing the same community again (loop). Numbers on arrows are estimates of the probability of taking a specific path.

that remaining in a stable climax community may be preferable to undergoing changes in community structure, as follows. Subjects with mild disease (Fig. 6B) were observed in three discrete states: Pa.M1, Str.D, and Oth.D. Of these, Pa.M1 is most like what has been described as a CF climax community, and Str.D is most like what has been described as an attack community. Stability in a Markov chain is defined as a transition to the current state and is denoted by a loop back to the same node. Figure 6B shows that subjects in the Pa.M1 state had the highest probability of remaining in the same state (0.73), consistent with concept Pa.M1 being a climax community. Subjects with mild disease (Fig. 6B) were significantly more likely to remain in Pa.M1 than subjects with moderate/severe disease (red arrow in 6B: * indicates $P = 0.03$, Fisher's exact test, odds ratio 1.78), which is consistent with the climax-attack model's prediction that stability in a climax state protects lung function. Transitions from Pa.M1 to Str.D may represent transitions from a climax state to an attack state.

One reason subjects with mild disease might appear more stable in this model is that they were more frequently observed, leaving less time to elapse during which a community might change in structure. However, greater stability in mild subjects cannot be attributed to an insufficient amount of time having passed between measurements to observe a change, because their average time between measurements (0.64 years) was somewhat longer than the average time between measurements in moderate/severe subjects (0.55 years), though this difference did not achieve significance ($P$ value = 0.16, Welch's $t$ test).

Our Markov chain analysis also provides support for other aspects of the climax-attack model, though these relationships did not achieve statistical significance. Communities that were dominated by *Streptococcus* (Str.D) were most likely to succeed Oth.D communities. This suggests that Str.D communities in sputum, likely descendants of pioneers that migrated from the mouth, may displace climax communities dominated by *Pseudomonas* or other traditional CF pathogens such as *Burkholderia* or *Achromobacter*. This finding is therefore generally consistent with both the climax-attack model and the island biogeography model (50).

It is interesting that the Pa.D community was detected in subjects with severe disease (Fig. 6C) but not in subjects with mild disease (Fig. 6B), and that the Pa.D state was not stable. This observation suggests that in our system, mixed communities with high levels of *Pseudomonas* behave more like climax communities than communities with the highest levels of *Pseudomonas*. Since the Pa.D community was rarely

observed, it is not possible to make statistical statements about its possible association with disease state or disease aggressiveness. However, it is possible that our data reflect a biological constraint; perhaps mild disease is incompatible with very high levels of *Pseudomonas*, even for brief periods.

## DISCUSSION

We present two major findings in this report. First, the previously reported climax-attack model postulates that an attack community composed of genera like *Streptococcus*, *Veillonella*, and *Porphyromonas* perturbs a climax community dominated by *Pseudomonas* and *Staphylococcus* (28), resulting in a metabolic shift toward fermentation and decreasing pH. These conditions are more favorable to fermentative bacteria like *Prevotella*, *Veillonella*, *Streptococcus*, *Rothia*, and *Granulicatella* (28). Our observation that mild disease is associated with community stability, and that stability is associated with a *Pseudomonas*-dominated community, are therefore consistent with the climax-attack model of CF.

Second, we have shown that a relatively small number of CF microbiome types explain patient differences in lung function, and that they explain these differences much better than single variables such as the relative abundance of *Pseudomonas*, age, or Simpson diversity. This finding is important because developing interventions to treat the community as the pathogenic unit will require a reasonable number of experimentally tractable model communities. Without such models, we would be faced with a dizzying array of patient differences. Fortunately, the airway community in a pwCF over multiple samplings generally looks more like "self" than any other patient when using relative abundance of bacteria as a metric of what defines a community, thereby increasing the likelihood of identifying an *in vitro* model that might help design more effective treatment regimens for a pwCF that has a similar community composition. Our findings raise the question of what combinations of abundant microbes (*Pseudomonas*, *Streptococcus*, and *Staphylococcus*) and less abundant community members (i.e., anaerobic, fermentative organisms) really matter. Does it matter if a patient has 90% *Pseudomonas* versus 80% or 70%? Intriguingly, *Pseudomonas*, often listed as the quintessential CF pathogen, is highly abundant in long-lived people with CF, i.e., it is an aspect of a "survivor phenotype" (51). Alternatively, does it matter if the dominant microbe in a community is *Streptococcus* rather than *Pseudomonas*, or if the minor players are *Prevotella* versus *Veillonella* versus *Gemella*? Although unsupervised machine learning cannot answer these questions with precision, we have demonstrated that as few as five communities may represent a useful starting point, as pwCF with a particular community type experience significantly similar lung function as measured by ppFEV$_1$.

There are several critical factors and caveats to consider in light of the interpretations we make here. First, although a gap statistic suggests five communities may be optimal for the cross-sectional data set used here, one should avoid construing "five" as the number of distinct populations in the CF community. Children younger than eight were not included in this study, and it is possible that broad sampling of young pwCF would have enabled us to identify additional clusters. In the same vein, although cluster 3 (Oth.D) contains subjects that were dominated by genera other than *Pseudomonas*, it is understood that broader sampling of this group would create additional clusters for each genus in cluster 3. Nonetheless, some of the findings here mirror the conclusions from previous studies with smaller data sets (12, 15, 38). Second, this study includes both cross-sectional and longitudinal observations of airway microbiomes and associated clinical parameters, enabling us to identify associations between them. However, this type of study cannot prove causation. Indeed, our interpretation of the data here should rightly be viewed as hypotheses to be tested either via additional clinical studies or in the laboratory. Third, and related to the previous point, the most useful aspect of this work is that it does set a framework for hypothesis testing; for example, can we build the communities we observe here *in vitro* and assess their function (i.e., are some clusters of microbes more/less cytotoxic or more/less

recalcitrant to antibiotics)? Alternatively, are similar communities associated with positive/negative health outcomes in other cohorts of pwCF or in other disease settings? For example, a recent report by Frey et al. (29) concludes that airway diversity is a biomarker of lung disease severity that links dysbiosis to lung function decline in pwCF. They observed a negative correlation between inflammatory cytokine secretion and airway community diversity, suggesting that more diverse communities are for some reason more salubrious than less diverse communities (a common but unexplained finding across many microbiome studies). Using a slightly different unsupervised approach to identify CF airway communities (i.e., hierarchical clustering of Unifrac distances), these investigators found that high diversity maps to an oropharyngeal-like community typically found in younger people. The authors carefully avoid making the claim that bacterial diversity reduces inflammation, although others have reported this association (43) or found a positive association between higher lung function and increased bacterial diversity (14, 17). The diverse, oropharyngeal-like community identified by Frey and colleagues is characterized by anaerobes like *Prevotella* and *Veillonella* and aerobic commensals, including *Neisseria* and *Streptococcus*, making it very similar to the *Streptococcus*-dominated community (Str.SD) identified in this report (Fig. 3). Of note, our Str.D community was also associated with higher lung function (Fig. 4A) and relative youth (Fig. 4B) in pwCF. Thus, orthogonal approaches appear to converge on similar conclusions as to community composition associated with less severe health outcomes.

The findings here and reported by Frey et al. (29) and others (14, 43, 46) raise the question of whether remodeling CF airway communities to make them more diverse and more similar to the oropharyngeal communities found in younger people with higher lung function, if possible, might improve outcomes in CF. There are several reasons to question the wisdom of attempting to turn back the hands of time in older subjects and enhance the abundance of oropharyngeal flora in their airways. First, as described above, it is possible that the introduction of diversity into an established community (for example, one dominated by *Pseudomonas*) would destabilize the community and lead to enhanced inflammation, as suggested by the climax-attack model of CF (49). Second, as demonstrated by Muhlebach et al. (52), inflammation increases when pediatric pwCF are first colonized by oropharyngeal flora; although this finding should not be surprising, it is worth pointing out that the best airway community is probably no community at all. Third, even in adults, increased diversity is not uniformly benign. Studies by Raghuvanshi et al. (24) show a linear increase in the microbiome alpha-diversity and in the log ratio of anaerobes to pathogens 2 weeks before a pulmonary exacerbation, confirming similar observations by Carmody et al. (18) showing that bacterial community diversity and the relative abundance of anaerobes increased during exacerbations, i.e., before antibiotic treatment. Others have shown that treating adults with antibiotics primarily affects taxa other than classic CF pathogens (53, 54), and that antibiotics reduce the abundance of *Streptococcus*, *Prevotella*, and *Veillonella*, but not *Pseudomonas*, the ostensible target of treatment. Collectively, then, the results cited support the somewhat surprising conclusion that oropharyngeal flora are associated with worse airway disease, and that treatments work because they reduce the abundance of oropharyngeal genera.

Finally, as mentioned above, we present just one approach for analyzing the extant microbial community data associated with CF airway infections, but an approach that has both statistical support and practical utility. That is, it is likely that only controlled experiments using model communities in a laboratory setting or in animals can unravel the mechanistic role played by oropharyngeal-like flora in an established adult microbiome (14, 25, 27, 43), thereby enhancing our understanding of the functional consequences of community composition. That is, we have identified 5 communities comprising 10 genera, which present an experimentally tractable number of communities and microbes. As another example, Crabbe et al. (55) recently reported using a nine-member community for *in vitro* studies. Going forward, this and other studies can help

inform the nature of laboratory model communities used to probe the hypotheses we and others have developed by analyzing clinical data sets.

## MATERIALS AND METHODS

**Data processing.** Publicly available 16S rRNA gene amplicon library sequence reads and associated metadata from recent studies by Carmody et al. (PRJNA423040) (18) and Cuthbertson et al. (PRJEB30646) (17) were downloaded from Bioproject (https://www.ncbi.nlm.nih.gov/bioproject/). Reads in fastq format were mapped to unique bacterial genera using QIME2 (56).

**Cross-sectional data set.** We assembled a cross-sectional data set by combining baseline data (i.e., not exacerbating, not receiving treatment for exacerbation) set from 167 unique pwCF, composed of 84 from the Cuthbertson et al. study (17) and 83 from the Carmody study (18). In those instances where more than one baseline sample was available from a given subject, the first sample taken was used, together with metadata (e.g., ppFEV$_1$). The two data sets were similar but not identical in terms of age and lung function. The 84 subjects included in the cross-sectional data set from the Carmody et al. study (18) were slightly younger, with a mean age of $25.6 \pm 10.0$ standard deviation (SD) compared to a mean age of $30.27 \pm 10.9$ SD. Operational taxonomic unit (OTU)-level mapping and metadata for the two cross-sectional data sets (17, 18) is provided in Table S1 and Table S2 in the supplemental material.

Samples in the two studies were processed using somewhat different protocols. The Carmody protocol involved breaking down bacterial cell walls by bead beating, DNA extraction using a MagNA Pure nucleic acid purification platform, amplification of the V3, V4, and V5 hypervariable regions, and Roche 454 pyrosequencing. The Cuthbertson protocol included pretreatment with propidium monoazide prior to DNA extraction to prevent sequencing of reads from dead bacteria. This step alone should create systematic differences between the two data sets, even if one assumes identical starting material. Two other differences between the two protocols would also be predicted to add to systematic differences between the two data sets, namely, amplification of different hypervariable regions (V1 to V2) and sequencing on the Illumina MiSeq platform using V3 chemistry. Not surprisingly, a permutational multivariate analysis of variance (PERMANOVA) of Bray-Curtis dissimilarity between the two data sets estimates that about 18% of the variability in the cross-sectional data set is explained by the study from with a sample originated ($r^2 = 0.181$, $P < 0.0001$). Since subjects in the two studies were of slightly different ages and were treated in different health care systems (essentially, United States in one case and Europe in the other), it is likely that systematic differences between communities reflect both biological and technical differences. Although it is not possible to assess the relative contribution of technically and biologically derived variability in our cross-sectional data, we can say that more than 82% of the variability in these data arose from nontechnical sources. In short, although technical differences probably add noise to this meta-analysis of microbiome data, large biologically driven signals should be detectable, as others have shown (45).

**Longitudinal data set.** The longitudinal data set was a subset of samples from the Carmody et al. (18) study, specifically, 346 16S rRNA gene amplicon data sets from 24 subjects that were observed at least 10 times. Age at time of sampling ranged from 11 to 54, with median of 27 and an interquartile range of 8. Lung function (ppFEV$_1$) of subjects in the longitudinal data set ranged from 9% to 123%, with a median of 50 and an interquartile range of 36. Longitudinal subjects are more fully described in Table 2. OTU level mapping and metadata for the longitudinal data set (18) is provided in Table S3. Our longitudinal data analysis contrasted subjects with mild disease with those classified as having moderate/severe disease, classifications that were annotated by Carmody et al. (18) using a method developed by Schluchter et al. (57). This classification system uses annual changes in lung function to predict what a subject's lung function would be at 20 years of age. For example, a person whose rate of decline during childhood predicts a ppFEV$_1$ of 35% by age 20 would be classified as having severe disease.

**Statistical analyses.** Data analysis was performed in R 4.0.1 (https://www.R-project.org/). Bray-Curtis dissimilarity was calculated using ecodist (58). Simpson diversity was calculated with vegan 2.5-7 (59). Clustering was performed using kmeans in base R and clusters were visualized and gap analysis was performed using factoextra 1.0.7 (60). Random forests were generated using randomForest 4.6-14 (61). Discrete Time Markov chains were determined using the markovchain package 0.8.5-4 (62).

## SUPPLEMENTAL MATERIAL

Supplemental material is available online only.

**SUPPLEMENTAL FILE 1**, CSV file, 0.1 MB.
**SUPPLEMENTAL FILE 2**, CSV file, 0.1 MB.
**SUPPLEMENTAL FILE 3**, CSV file, 0.3 MB.

## ACKNOWLEDGMENTS

Support for these studies was provided by the Cystic Fibrosis Foundation (CFF STANTO19R0 and STANTO19GO) and NIH (R01-HL151385, R01-AI155424).

We thank the NIH P30-DK117469 for facilitating analysis of publicly available data sets and for advice to Dartmouth colleagues.

We thank John LiPuma for access to his team's data and many helpful insights about this project.

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
