## [Reviewer comments · Microbiology Spectrum]

Microbiology Spectrum

Mild CF Lung Disease is Associated with Bacterial Community Stability

Thomas Hampton, Devin Thomas, Christopher van der Gast, George O'Toole, and Bruce Stanton

Corresponding Author(s): George O'Toole, Geisel School of Medicine at Dartmouth

Review Timeline:

Submission Date:	April 13, 2021
Editorial Decision:	May 20, 2021
Revision Received:	June 8, 2021
Accepted:	June 9, 2021

Editor: Amanda Oglesby

Reviewer(s): The reviewers have opted to remain anonymous.

Transaction Report:

DOI: <https://doi.org/10.1128/Spectrum.00029-21>

May 20, 2021

Dr. George A O'Toole
Dartmouth College
Biological Sciences
The Class of 1978 Life Sciences Center
78 College Street
Hanover, NH 03755

Re: Spectrum00029-21 (Mild CF Lung Disease is Associated with Bacterial Community Stability)

Dear Dr. George A O'Toole:

Thank you for submitting your manuscript to Microbiology Spectrum. As you will see the reviewers support publication of a revised paper. I do not believe additional analysis will be required to address these comments; rather, the reviewers have requested clarification of the rationale for the current methods. Please revise the paper along the lines suggested by the reviewers.

When submitting the revised version of your paper, please provide (1) point-by-point responses to the issues raised by the reviewers as file type "Response to Reviewers," not in your cover letter, and (2) a PDF file that indicates the changes from the original submission (by highlighting or underlining the changes) as file type "Marked Up Manuscript - For Review Only". Please use this link to submit your revised manuscript - we strongly recommend that you submit your paper within the next 60 days or reach out to me. Detailed information on submitting your revised paper are below.

Link Not Available

Sincerely,

Amanda Oglesby

Journals Department
Reviewer comments:

Reviewer #1 (Comments for the Author):

Summary: In this meta-analysis, Hampton and colleagues examine two previously published 16S amplicon microbiome sequencing datasets from sputum of 167 people with CF (pwCF) in the US and Europe. There are two main findings. First, the authors reduce the total combined dataset into a cross-sectional dataset and use an unsupervised clustering approach to identify 5 microbiome bacterial community types that explain more variability in lung function in the cohort compared to select other predictors, including Simpson diversity or dominance by *Pseudomonas* spp. Second, using another subset of the combined dataset (longitudinal samples from 24 individuals in the Carmody et al. study, previously classified as having either mild or moderate/severe disease), the authors use a random forest classifier to assign the longitudinal samples into one of the 5 microbiome community types, then assess the stability over time using by presenting a few statistics and graphing as Discrete Time Markov chains. From these analyses with the longitudinal dataset, they found that individuals with mild disease tended to be more stable over time, remaining in a community type that was on average largely *Pseudomonas* spp., but contained other taxa as well (Cluster 4/Pa.M1). The authors conclude that this meta-analysis provides some support for the Climax-Attack Model of CF microbial community dynamics during periods of clinical stability versus pulmonary exacerbation, as well as suggesting that instability may predict disease progression and that the 5 microbial community types may inform future in vitro mechanistic studies of polymicrobial interactions in CF.

Critique: This is a nice paper and the authors do a good job of tempering their interpretations, rather than overstating their findings. The ability of the 5 community types to explain a large amount of variability in these two cohorts' ppFEV1 is important and this approach will be a helpful addition to the CF microbiome literature. I also appreciated the degree to which the authors tried to put their findings into context with the current literature in the field. My major comments are mostly related to a couple of larger points that I think need to be clarified or better discussed, including a potential issue stemming from differences between sequencing approaches in the Cuthbertson and Carmody studies. I also included minor comments that will help to strengthen the paper. Overall, I am enthusiastic about this study and appreciate the thoughtful, measured interpretations provided by the authors.

Major comments:

(1) While this is a meta-analysis of previously published datasets, there were a couple of instances where I thought the authors could include some information regarding methods from the original publications, rather than relying on readers to dig up the info:

1a) Comparing sample collection/sequencing methods between the two datasets: A major concern when conducting a meta-analysis of microbiome datasets from different studies is that differences in sample processing and/or sequencing methods between studies may drive differences in, for example, alpha diversity or particular taxa detected between cohorts (e.g. PMID: 23861384). While the authors somewhat address this with the results presented in Table 2, they should also include a little more information in the methods section (around line 120), including: Which primers/which regions of the 16S gene were sequenced in each study (it looks like Carmody amplified V3-V4 and Cuthbertson amplified V1-V2)? What sequencing platform was used (looks like Carmody used 454 technology and Cuthbertson used Illumina)? Were there any notable differences in sample processing/extraction methods? Does the source study (Carmody versus Cuthbertson) significantly explain any differences between samples (e.g. by PERMANOVA)? Do the authors expect that any

of the differences between samples from the two studies described in Table 2 are due to technical differences between the studies?

1b) Since the designations from the Carmody study of "mild" versus "moderate/severe" disease were used in several figures and are important to overall findings, the authors should briefly explain in the methods how these designations were assigned (e.g. by a physician, based in some way on FEV1 or class of CFTR mutation, self-scored by the study participant, etc).

(2) In my opinion, the paper is missing a discussion as to how the Markov chain networks do or do not relate to the Climax-Attack model. Central to the Climax-Attack model is the transition between distinct climax and attack communities in association with pulmonary exacerbations, and the networks show community transitions.

2a) Can the authors speculate as to whether any of the community transitions and their associated probabilities in Figures 6B and 6C represent transitions between climax and attack communities?

2b) The authors point out with a red arrow in Figure 6B that repeated classification in the Pa.M1 community is significantly more likely in people with mild disease, compared to severe disease. Can the authors speculate whether/how this stable membership in Pa.M1 may relate to the Climax-Attack model or other CF microbial ecology models (e.g. those recently reviewed in DOI: 10.1128/mSystems.00809-20)? Along these lines, were any other probabilities of community transitions significantly different between Figures 6B and 6C?

2c) Why is the community Pa.D missing from the mild disease cohort? Is this community only observed in people with severe disease or is this more likely an issue related to a small sample size/comparatively few individuals' visits in the longitudinal cohort being classified as Pa.D communities?

Minor comments:

(3) Lines 157-159 and elsewhere in the manuscript, the authors switch between Bray-Curtis "similarity" and "dissimilarity". This is a little confusing. I am guessing that the statistic that was estimated by the authors was the Bray-Curtis dissimilarity index, so I suggest sticking to the term "dissimilarity".

(4) On line 221, the authors present adjusted r^2 values from linear models of ppFEV1 as a function of cluster membership. Were any covariates such as age or sex included in these models? For example, on line 164, the authors note that subjects in the combined cohort ranged in age from 8 to 69. Since lung function tends to decline and relative abundances of taxa in sputum differ as pwCF age, the authors should describe whether/how they controlled for age and any other common covariates (e.g. sex, CFTR mutation) in their models.

(5) Lines 246-250 and Figure 2: Does cluster membership plus age or any of the other variables tested separately improve the ability of the linear model to predict ppFEV1? It isn't clear to me that this was tested. However, if it improves the model, it could suggest that knowing the microbiome community type in the context of knowing the age of the patient, for example, could be the most informative in estimating a person's lung function.

(6) Lines 262-263: The sentence beginning "Choosing the top 10 genera..." was confusing to me. Is this saying that, for example, for the genus *Pseudomonas*, there were multiple OTU's/ASV's, but that the OTU/ASV that was among the top 10 contained most of the reads that belong to *Pseudomonas*?

(7) Lines 302-304: The sentence beginning "We therefore complemented..." is confusing. Does this mean that for every sample, the OTU/ASV that was highest in relative abundance was noted and

then, the top 10 genera were ranked from this list of one top OTU/ASV per sample?

(8) Figure 3C: I would recommend adding the mnemonics/cluster names along the bottom, in addition to "Cluster 1, Cluster 2, etc" just so the reader doesn't need to flip around to find them.

(9) Figures 4AB show the association of age and ppFEV1 with community membership. Controlling for age, does ppFEV1 still differ between communities in Figure 4A?

(10) Lines 377-378: "We did not observe large differences in community types between subjects with mild versus moderate/severe disease..." I noticed that Community Pa.D is only observed in the moderate/severe group (e.g. based on coloring in Figure 5AB). This seems important to me, but the authors don't mention it. Can the authors speculate as to why this community absent from the mild disease group?

(11) The networks in figure 6BC need more explanation of which numbers represent which probabilities, since there are typically two numbers per edge. I see that the numbers closest to the nodes tend to add up to 1, so those represent the probability that a microbiome sample will be classified as the group denoted by the closest node, given they were previously classified in whatever the other node was along that edge? Why do the numbers along the edges leading into the Pa.D node on Figure 6D equal greater than 1 ($1 + 0.03$)? Is there a 0 missing from that edge?

(12) Did the authors check to see that the difference in "stability" between the mild and moderate/severe individuals is not due to the length of time between samples? For example, in Figures 6BC, was the amount of time between sampling for the dates when an individual does not change microbiome groups (i.e. the samples represented by an edge that points back into the same node) less than the amount of time between sampling for the times when an individual switched microbiome groups?

Reviewer #2 (Comments for the Author):

This manuscript by Hampton and colleagues is a kind of meta-analysis of two preexisting datasets to identify whether or not there is an association between community stability and lung function in CF. The manuscript aims to generate hypothesis, which it does, and it is clearly written. A few minor concerns to address:

1. The methodology used to generate the two reference datasets that were combined is quite different: the Carmody set is based on 454 pyrosequencing and the Cuthbertson on Illumina sequencing (among other significant differences, including a decade difference in timeframe of collection). Although this does not affect the results per se the authors would do well to discuss limitations from combining the varying approaches and how it might affect the interpretation.
2. How much of the apparent stability-to-lung function association (for the whole study) can be attributed to Str.D community, which was really only present at a single site?
3. The authors should explain a little more why they chose 5 communities. Yes, 6 had a lower r -squared, but 7, 8, or 9 were higher. (And 8 or 9 had a higher gap statistic, too). This is NOT to suggest a re-analysis of all the data with more groups, but instead for more clarity on the choice.
4. The Figure 1B colors would be impossible to differentiate for a person who is colorblind. Some other marker to delineate the differences would be helpful.

Staff Comments:

Preparing Revision Guidelines

For complete guidelines on revision requirements, please see the Instructions to Authors at [link to page]. **Submissions of a paper that does not conform to Microbiology Spectrum guidelines will delay acceptance of your manuscript.**

Please return the manuscript within 60 days; if you cannot complete the modification within this time period, please contact me. If you do not wish to modify the manuscript and prefer to submit it to another journal, please notify me of your decision immediately so that the manuscript may be formally withdrawn from consideration by Microbiology Spectrum.

If you would like to submit an image for consideration as the Featured Image for an issue, please contact Spectrum staff.

We appreciate the reviewers' careful reading of our work. Addressing their comments has improved this manuscript.

Original reviewer comments are included for reference (indicated by >>COMMENTS); our responses are indicated (RESPONSES). Line numbers at which text changes begin refers to line numbers in the revised manuscript. Our updated text is indented.

Reviewer #1 (Comments for the Author):

>>COMMENT: Summary: In this meta-analysis, Hampton and colleagues examine two previously published 16S amplicon microbiome sequencing datasets from sputum of 167 people with CF (pwCF) in the US and Europe. There are two main findings. First, the authors reduce the total combined dataset into a cross-sectional dataset and use an unsupervised clustering approach to identify 5 microbiome bacterial community types that explain more variability in lung function in the cohort compared to select other predictors, including Simpson diversity or dominance by *Pseudomonas* spp. Second, using another subset of the combined dataset (longitudinal samples from 24 individuals in the Carmody et al. study, previously classified as having either mild or moderate/severe disease), the authors use a random forest classifier to assign the longitudinal samples into one of the 5 microbiome community types, then assess the stability over time using by presenting a few statistics and graphing as Discrete Time Markov chains. From these analyses with the longitudinal dataset, they found that individuals with mild disease tended to be more stable over time, remaining in a community type that was on average largely *Pseudomonas* spp., but contained other taxa as well (Cluster 4/Pa.M1). The authors conclude that this meta-analysis provides some support for the Climax-Attack Model of CF microbial community dynamics during periods of clinical stability versus pulmonary exacerbation, as well as suggesting that instability may predict disease progression and that the 5 microbial community types may inform future in vitro mechanistic studies of polymicrobial interactions in CF.

RESPONSE: We appreciate this reviewer's close reading of our work.

>>COMMENT: Critique: This is a nice paper and the authors do a good job of tempering their interpretations, rather than overstating their findings. The ability of the 5 community types to explain a large amount of variability in these two cohorts' ppFEV1 is important and this approach will be a helpful addition to the CF microbiome literature. I also appreciated the degree to which the authors tried to put their findings into context with the current literature in the field. My major comments are mostly related to a couple of larger points that I think need to be clarified or better discussed, including a potential issue stemming from differences between sequencing approaches in the Cuthbertson and Carmody studies. I also included minor comments that will help to strengthen the paper. Overall, I am enthusiastic about this study and appreciate the thoughtful, measured interpretations provided by the authors.

RESPONSE: We thank the reviewer for their support of our work. As recommended, we have included a new section that discusses potential issues stemming from differences between sequencing approaches in the Cuthbertson and Carmody studies beginning on line 133:

Samples in the two studies were processed using somewhat different protocols. The Carmody protocol involved breaking down bacterial cell walls by bead beating, DNA extraction using a MagNA Pure nucleic acid purification platform, amplification of the V3, V4, and V5 hypervariable regions, and Roche 454 pyrosequencing. The Cuthbertson protocol included pre-treatment with propidium monoazide prior to DNA extraction to prevent sequencing of reads from dead bacteria. This step alone should create systematic differences between the two data sets, even

if one assumes identical starting material. Two other differences between the two protocols would also be predicted to add to systematic differences between the two data sets, namely, amplification of different hypervariable regions (V1-V2) and sequencing on the Illumina MiSeq platform using V3 chemistry. Not surprisingly, a PERMANOVA analysis of Bray-Curtis dissimilarity between the two data sets estimates that about 18% of the variability in the cross-sectional data set is explained by the study from which a sample originated ($r^2 = 0.181$, $P < .0001$). Since subjects in the two studies were of slightly different ages and were treated in different health care systems (essentially, United States in one case and Europe in the other) it is likely that systematic differences between communities reflect both biological and technical differences. Although it is not possible to assess the relative contribution of technically and biologically derived variability in our cross-sectional data, we can say that more than 82% of the variability in these data arose from non-technical sources. In short, although technical differences probably add noise to this meta-analysis of microbiome data, large biologically driven signals should be detectable, as others have shown (35).

Major comments:

>>COMMENT: (1) While this is a meta-analysis of previously published datasets, there were a couple of instances where I thought the authors could include some information regarding methods from the original publications, rather than relying on readers to dig up the info.

RESPONSE: We thank the reviewer for this suggestion and have described the methods used in the two studies and our assessment of how these differences might affect our findings. This new text is below and begins on line 133:

Samples in the two studies were processed using somewhat different protocols. The Carmody protocol involved breaking down bacterial cell walls by bead beating, DNA extraction using a MagNA Pure nucleic acid purification platform, amplification of the V3, V4, and V5 hypervariable regions, and Roche 454 pyrosequencing. The Cuthbertson protocol included pre-treatment with propidium monoazide prior to DNA extraction to prevent sequencing of reads from dead bacteria. This step alone should create systematic differences between the two data sets, even if one assumes identical starting material. Two other differences between the two protocols would also be predicted to add to systematic differences between the two data sets, namely, amplification of different hypervariable regions (V1-V2) and sequencing on the Illumina MiSeq platform using V3 chemistry. Not surprisingly, a PERMANOVA analysis of Bray-Curtis dissimilarity between the two data sets estimates that about 18% of the variability in the cross-sectional data set is explained by the study from which a sample originated ($r^2 = 0.181$, $P < .0001$). Since subjects in the two studies were of slightly different ages and were treated in different health care systems (essentially, United States in one case and Europe in the other) it is likely that systematic differences between communities reflect both biological and technical differences. Although it is not possible to assess the relative contribution of technically and biologically derived variability in our cross-sectional data, we can say that more than 82% of the variability in these data arose from non-technical sources. In short, although technical differences probably add noise to this meta-analysis of microbiome data, large biologically driven signals should be detectable, as others have shown (35).

>>COMMENT: A major concern when conducting a meta-analysis of microbiome datasets from different studies is that differences in sample processing and/or sequencing methods between studies may drive differences in, for example, alpha diversity or particular taxa detected between cohorts (e.g. PMID: 23861384). While the authors somewhat address this with the results presented in Table 2, they should also include a little more information in the methods section (around line 120), including: Which primers/which regions of the 16S gene were sequenced in each study (it looks like Carmody amplified V3-V4 and Cuthbertson amplified V1-V2)? What sequencing platform was used (looks like Carmody used 454 technology and Cuthbertson used Illumina)?

RESPONSE: We now include, beginning on line 133, details such as primers, variable regions, extraction methods and platforms used in each study:

Samples in the two studies were processed using somewhat different protocols. The Carmody protocol involved breaking down bacterial cell walls by bead beating, DNA extraction using a MagNA Pure nucleic acid purification platform, amplification of the V3, V4, and V5 hypervariable regions, and Roche 454 pyrosequencing. The Cuthbertson protocol included pre-treatment with propidium monoazide prior to DNA extraction to prevent sequencing of reads from dead bacteria. This step alone should create systematic differences between the two data sets, even

if one assumes identical starting material. Two other differences between the two protocols would also be predicted to add to systematic differences between the two data sets, namely, amplification of different hypervariable regions (V1-V2) and sequencing on the Illumina MiSeq platform using V3 chemistry. Not surprisingly, a PERMANOVA analysis of Bray-Curtis dissimilarity between the two data sets estimates that about 18% of the variability in the cross-sectional data set is explained by the study from which a sample originated ($r^2 = 0.181$, $P < .0001$). Since subjects in the two studies were of slightly different ages and were treated in different health care systems (essentially, United States in one case and Europe in the other) it is likely that systematic differences between communities reflect both biological and technical differences. Although it is not possible to assess the relative contribution of technically and biologically derived variability in our cross-sectional data, we can say that more than 82% of the variability in these data arose from non-technical sources. In short, although technical differences probably add noise to this meta-analysis of microbiome data, large biologically driven signals should be detectable, as others have shown (35).

>>COMMENT: Does the source study (Carmody versus Cuthbertson) significantly explain any differences between samples (e.g. by PERMANOVA)?

RESPONSE: Yes, we have updated our manuscript to state that source study explained about 18% of variability, and this was statistically significant, as described beginning on line 139:

Two other differences between the two protocols would also be predicted to add to systematic differences between the two data sets, namely, amplification of different hypervariable regions (V1-V2) and sequencing on the Illumina MiSeq platform using V3 chemistry. Not surprisingly, a PERMANOVA analysis of Bray-Curtis dissimilarity between the two data sets estimates that about 18% of the variability in the cross-sectional data set is explained by the study from which a sample originated ($r^2 = 0.181$, $P < .0001$).

>>COMMENT: Do the authors expect that any of the differences between samples from the two studies described in Table 2 are due to technical differences between the studies?

RESPONSE: Yes. We have updated our manuscript to clarify that both technical biological sources of variation probably contribute to the observed compositional differences between studies, beginning on line 298:

It is likely that both technical and clinical differences contribute to compositional differences observed between studies shown in Table 2. On the one hand, it is well established that CF practices and outcomes differ by region, both within and between continents (51), suggesting that CF airway microbiomes may be systematically different by region. On the other, it is well established that technical differences can cause apparent differences in community composition (35). A future study of regional CF airway microbiome differences, one in which all samples were processed identically, would be required to determine whether microbiomes truly cluster by geographic location.

>>COMMENT: 1b) Since the designations from the Carmody study of "mild" versus "moderate/severe" disease were used in several figures and are important to overall findings, the authors should briefly explain in the methods how these designations were assigned (e.g., by a physician, based in some way on FEV1 or class of CFTR mutation, self-scored by the study participant, etc).

RESPONSE: We have added a reference and description of how Carmody et. al. identified subjects with severe disease , beginning on line 160:

Our longitudinal data analysis contrasted subjects with mild disease with those classified as moderate/severe disease, classifications that were annotated by Carmody et al. (18) using a method developed by Schluchter et. al. (36). This classification system uses annual changes in lung function to predict what a subject's lung function would be at 20 years of age. For example, a person whose rate of decline during childhood predicts a ppFEV₁ of 35% by age 20 would be classified as having severe disease.

>>COMMENT: (2) In my opinion, the paper is missing a discussion as to how the Markov chain networks do or do not relate to the Climax-Attack model. Central to the Climax-Attack model is the transition between distinct climax and attack communities in association with pulmonary exacerbations, and the networks show community transitions.

RESPONSE: We have made the ways in which our Markov chain analysis does and does not relate to the Climax-Attack Model more explicit as follows. We observe that mild disease was associated fewer transitions out of the community type most similar in composition to previously described climax community which is consistent with predictions made by the Climax-Attack Model, beginning at line 406:

Nonetheless, the dynamic community structure detected by Markov chains is broadly consistent with cyclical behavior predicted by the Climax-Attack Model, and the proposition that remaining in a stable climax community may be preferable to undergoing changes in community structure, as follows.

RESPONSE: On the other hand, the community type most similar to an attack community was not always associated with a clinically observed exacerbation which would be predicted by the Climax-Attack Model, as described beginning at line 402:

In addition, the Str.D community, which would be most similar to previously described attack communities, did not significantly predict exacerbation state, as shown in Figure 5B.

>>COMMENT: 2a) Can the authors speculate as to whether any of the community transitions and their associated probabilities in Figures 6B and 6C represent transitions between climax and attack communities?

RESPONSE: Yes. We have clarified that the Pa.M1 and Str.D communities are most similar to climax and attack communities, as described beginning on line 410:

Of these, Pa.M1 is most like what has been described as a CF climax community, and Str.D is most like what has been described as an attack community.

RESPONSE: We also clarify that the increased probability of remaining in the climax community among subjects with mild disease is consistent with the hypothesis that transitions between Pa.M1 to Str.D may represent a transition from a climax community to an attack community on line 418:

Transitions from Pa.M1 to Str.D may represent transitions from a climax state to an attack state.

>>COMMENT: 2b) The authors point out with a red arrow in Figure 6B that repeated classification in the Pa.M1 community is significantly more likely in people with mild disease, compared to severe disease. Can the authors speculate whether/how this stable membership in Pa.M1 may relate to the Climax-Attack model or other CF microbial ecology models (e.g. those recently reviewed in DOI: 10.1128/mSystems.00809-20)?

RESPONSE: In addition to making the association between stable membership in Pa.M1 and the Climax-Attack model, we note that our results are also consistent with the Island Biogeography Model on line 431:

This suggests that Str.D communities in sputum, likely descendants of pioneers that migrated from the mouth may displace climax communities dominated by *Pseudomonas* or other traditional CF pathogens such as *Burkholderia* or *Achromobacter*. This finding is therefore generally consistent with both the Climax-Attack Model and the Island Biogeography Model (57)

>>COMMENT: 2c) Why is the community Pa.D missing from the mild disease cohort?

RESPONSE: We have changed the manuscript to address the question of why Pa.D is rare beginning on line 437:

It is interesting that the Pa.D community was detected in subjects with severe disease (Figure 6C) but not in subjects with mild disease (Figure 6B) and that the Pa.D state was not stable. This observation suggests that in our system, mixed communities with high levels of *Pseudomonas* behave more like climax communities than communities with the highest levels of *Pseudomonas*. Since the Pa.D community was rarely observed, it is not possible to make statistical statements about its possible association with disease state or disease aggressiveness.

>>COMMENT: Is this community only observed in people with severe disease or is this more likely an issue related to a small sample size/comparatively few individuals' visits in the longitudinal cohort being classified as Pa.D communities?

RESPONSE: We note that we have too few examples of this community to make definitive statistical statements regarding its possible association with severe disease. However, we speculate that absence of the Pa.D community in subjects classified as having mild disease may reflect a biological true biological constraint, beginning on line 437:

It is interesting that the Pa.D community was detected in subjects with severe disease (Figure 6C) but not in subjects with mild disease (Figure 6B) and that the Pa.D state was not stable. This observation suggests that in our system, mixed communities with high levels of *Pseudomonas* behave more like climax communities than communities with the highest levels of *Pseudomonas*. Since the Pa.D community was rarely observed, it is not possible to make statistical statements

about its possible association with disease state or disease aggressiveness. However, it is possible that our data reflect a biological constraint: perhaps mild disease is incompatible with very high levels of *Pseudomonas*, even for brief periods.

>>COMMENT: Minor comments:

(3) Lines 157-159 and elsewhere in the manuscript, the authors switch between Bray-Curtis "similarity" and "dissimilarity". This is a little confusing. I am guessing that the statistic that was estimated by the authors was the Bray-Curtis dissimilarity index, so I suggest sticking to the term "dissimilarity".

RESPONSE: We now only refer to Bray-Curtis dissimilarity.

>>COMMENT: (4) On line 221, the authors present adjusted r^2 values from linear models of ppFEV1 as a function of cluster membership. Were any covariates such as age or sex included in these models?

RESPONSE: We now clearly state that linear models of ppFEV1 as a function of cluster membership do not include other covariates, beginning on line 259:

In contrast, membership in one of the 5 communities identified above explains 24% of patient variability in lung function, without including any other covariates.

>>COMMENT: For example, on line 164, the authors note that subjects in the combined cohort ranged in age from 8 to 69. Since lung function tends to decline and relative abundances of taxa in sputum differ as pwCF age, the authors should describe whether/how they controlled for age and any other common covariates (e.g. sex, CFTR mutation) in their models.

RESPONSE: We now clearly state that although sex and CFTR mutation were present in the van der Gast data set, these annotations were not part of the Carmody annotations, therefore these covariates were not included in our analysis, beginning on line 250:

Although sex and CFTR mutation were present in the van der Gast data set (17), these annotations were not part of the Carmody data set (18), therefore these covariates were not included in our analysis

RESPONSE: In addition, we now clearly state that linear models of ppFEV1 as a function of cluster membership do not include other covariates, beginning on line 259:

In contrast, membership in one of the 5 communities identified above explains 24% of patient variability in lung function, without including any other covariates.

>>COMMENT: (5) Lines 246-250 and Figure 2: Does cluster membership plus age or any of the other variables tested separately improve the ability of the linear model to predict ppFEV1? It isn't clear to me that this was tested. However, if it improves the model, it could suggest that knowing the microbiome community type in the context of knowing the age of the patient, for example, could be the most informative in estimating a person's lung function.

RESPONSE: We have followed up on this suggestion and assessed the extent to which the ability of the linear model to predict ppFEV1 based on cluster membership is improved by adding age or Simpson diversity to the model. Adding these covariates does not significantly improve predictive power, as described beginning at line 259:

In contrast, membership in one of the 5 communities identified above explains 24% of patient variability in lung function, without including any other covariates. Adding age or Simpson diversity to the community model did not significantly improve its predictive power. In other words, knowing which cluster a person's airway microbiome belongs to provides a better estimate of their lung function than knowing the relative abundance of *Pseudomonas* in their airway microbiome, the diversity of their community or their age (Figure 2).

>>COMMENT: (6) Lines 262-263: The sentence beginning "Choosing the top 10 genera..." was confusing to me. Is this saying that, for example, for the genus *Pseudomonas*, there were multiple OTU's/ASV's, but that the OTU/ASV that was among the top 10 contained most of the reads that belong to *Pseudomonas*?

RESPONSE: We thank the reviewer for pointing out the confusing nature of this statement, which has been removed and replaced by text that highlights our intended point that 10 genera do not account for 100% of relative abundance, beginning at line 268:

We identified the top 10 genera with the highest mean relative abundance across the 167 cross-sectional samples and averaged their relative abundance in each of the 5 clusters, as shown in Figure 3A. These 10 genera accounted for 87% of the relative abundance in Cluster 1 (Str.D), ...

>>COMMENT: (7) Lines 302-304: The sentence beginning "We therefore complemented..." is confusing. Does this mean that for every sample, the OTU/ASV that was highest in relative abundance was noted and then, the top 10 genera were ranked from this list of one top OTU/ASV per sample?

RESPONSE: We thank the reviewer for pointing out the confusing nature of this statement, which has been removed and replaced by text describing our intended point, that how averages do a poor job summarizing relative abundance in cluster 3 (Oth.D) beginning on line 311:

A sample-by-sample analysis of dominance revealed that *Haemophilus*, *Stenotrophomonas*, *Burkholderia*, *Prevotella* and *Achromobacter* all achieved high levels of dominance (relative abundance greater than 85%) in Cluster 3, but not in other clusters (Figure 3C). *Streptococcus* achieved dominance in Cluster 1, but not elsewhere.

>>COMMENT: (8) Figure 3C: I would recommend adding the mnemonics/cluster names along the bottom, in addition to "Cluster 1, Cluster 2, etc" just so the reader doesn't need to flip around to find them.

RESPONSE: We have made this change to Figure 3C.

>>COMMENT: (9) Figures 4AB show the association of age and ppFEV1 with community membership. Controlling for age, does ppFEV1 still differ between communities in Figure 4A?

RESPONSE: Controlling for age, ppFEV1 still differs significantly between communities, as indicated by new text starting at line 324:

Clusters differ significantly in ppFEV₁ after correcting for age in this model.

>>COMMENT: (10) Lines 377-378: "We did not observe large differences in community types between subjects with mild versus moderate/severe disease..." I noticed that Community Pa.D is only observed in the moderate/severe group (e.g. based on coloring in Figure 5AB). This seems important to me, but the authors don't mention it. Can the authors speculate as to why this community absent from the mild disease group?

RESPONSE: This is a great question. we speculate that absence of the Pa.D community in subjects classified as having mild disease may reflect a biological true biological constraint, beginning on line 437:

It is interesting that the Pa.D community was detected in subjects with severe disease (Figure 6C) but not in subjects with mild disease (Figure 6B) and that the Pa.D state was not stable. This observation suggests that in our system, mixed communities with high levels of *Pseudomonas* behave more like climax communities than communities with the highest levels of *Pseudomonas*. Since the Pa.D community was rarely observed, it is not possible to make statistical statements about its possible association with disease state or disease aggressiveness. However, it is possible that our data reflect a biological constraint: perhaps mild disease is incompatible with very high levels of *Pseudomonas*, even for brief periods.

>>COMMENT: (11) The networks in figure 6BC need more explanation of which numbers represent which probabilities, since there are typically two numbers per edge. I see that the numbers closest to the nodes tend to add up to 1, so those represent the probability that a microbiome sample will be classified as the group denoted by the closest node, given they were previously classified in whatever the other node was along that edge?

Why do the numbers along the edges leading into the Pa.D node on Figure 6D equal greater than 1 (1 + 0.03)? Is there a 0 missing from that edge?

RESPONSE: We thank the reviewer for pointing out this apparent inconsistency. We have added a zero to the figure to clarify this point.

>>COMMENT: (12) Did the authors check to see that the difference in "stability" between the mild and moderate/severe individuals is not due to the length of time between samples?

For example, in Figures 6BC, was the amount of time between sampling for the dates when an individual does not change microbiome groups (i.e. the samples represented by an edge that points back into the same node) less than the amount of time between sampling for the times when an individual switched microbiome groups?

RESPONSE: This is a great suggestion. We have added text to clarify that greater stability in mild subjects cannot be attributed to an insufficient amount of time having passed between measurements to observe a change, because their average time between measurements (0.64 years) was somewhat longer than average time between measurements in moderate/severe subjects (0.55 years) though this difference did not achieve significance (p -value = 0.16, Welch's t test). We have added text to this effect, beginning on line 421:

One reason subjects with mild disease might appear more stable in this model is that they were more frequently observed, leaving less time elapse during which a community might change in structure. However, greater stability in mild subjects cannot be attributed to an insufficient amount of time having passed between measurements to observe a change, because their average time between measurements (0.64 years) was somewhat longer than average time between measurements in moderate/severe subjects (0.55 years) though this difference did not achieve significance (p -value = 0.16, Welch's t test).

Reviewer #2 (Comments for the Author):

>>COMMENT: This manuscript by Hampton and colleagues is a kind of meta-analysis of two preexisting datasets to identify whether or not there is an association between community stability and lung function in CF. The manuscript aims to generate hypothesis, which it does, and it is clearly written. A few minor concerns to address:

1. The methodology used to generate the two reference datasets that were combined is quite different: the Carmody set is based on 454 pyrosequencing and the Cuthbertson on Illumina sequencing (among other significant differences, including a decade difference in timeframe of collection). Although this does not affect the results per se the authors would do well to discuss limitations from combining the varying approaches and how it might affect the interpretation.

RESPONSE: As recommended, we have included a new section that discusses potential issues stemming from differences between sequencing approaches in the Cuthbertson and Carmody studies, beginning on line 133:

Samples in the two studies were processed using somewhat different protocols. The Carmody protocol involved breaking down bacterial cell walls by bead beating, DNA extraction using a MagNA Pure nucleic acid purification platform, amplification of the V3, V4, and V5 hypervariable regions, and Roche 454 pyrosequencing. The Cuthbertson protocol included pre-treatment with propidium monoazide prior to DNA extraction to prevent sequencing of reads from dead bacteria. This step alone should create systematic differences between the two data sets, even if one assumes identical starting material. Two other differences between the two protocols would also be predicted to add to systematic differences between the two data sets, namely, amplification of different hypervariable regions (V1-V2) and sequencing on the Illumina MiSeq platform using V3 chemistry. Not surprisingly, a PERMANOVA analysis of Bray-Curtis dissimilarity between the two data sets estimates that about 18% of the variability in the cross-sectional data set is explained by the study from which a sample originated (r squared = 0.181, $P < .0001$). Since subjects in the two studies were of slightly different ages and were treated in different health care systems (essentially, United States in one case and Europe in the other) it is likely that systematic differences between communities reflect both biological and technical differences. Although it is not possible to assess the relative contribution of technically and biologically derived variability in our cross-sectional data, we can say that more than 82% of the variability in these data arose from

non-technical sources. In short, although technical differences probably add noise to this meta-analysis of microbiome data, large biologically driven signals should be detectable, as others have shown (35).

>>COMMENT: 2. How much of the apparent stability-to-lung function association (for the whole study) can be attributed to Str.D community, which was really only present at a single site?

RESPONSE: This is a great point, and we have highlighted the relationship between this community and the Carmody data beginning on line 270:

These 10 genera accounted for 87% of the relative abundance in Cluster 1 (Str.D), to which 46 samples belonged. This cluster was dominated by *Streptococcus* (brick red). It is noteworthy that this cluster was almost exclusively identified in the Carmody (18) data set, which contained a younger group of subjects.

RESPONSE: We have also estimated the overall variability attributable to differences between studies (18%) beginning on line 142, which includes all compositional differences attributable to study:

Not surprisingly, a PERMANOVA analysis of Bray-Curtis dissimilarity between the two data sets estimates that about 18% of the variability in the cross-sectional data set is explained by the study from with a sample originated ($r^2 = 0.181$, $P < .0001$).

RESPONSE: Importantly, Str.D is the putative attack community in our longitudinal analysis, and therefore plays a key role in stability, a point we have clarified on line 411:

Str.D is most like what has been described as an attack community.

>>COMMENT: 3. The authors should explain a little more why they chose 5 communities. Yes, 6 had a lower r-squared, but 7, 8, or 9 were higher. (And 8 or 9 had a higher gap statistic, too). This is NOT to suggest a re-analysis of all the data with more groups, but instead for more clarity on the choice.

RESPONSE: We have added text to address the reviewer's excellent question. Briefly, the algorithm is designed to identify the smallest k supported by the data, beginning on line 222:

The basis of this choice is driven by the break in the data after $k=5$. Specifically, before $k=5$, the gap statistic increases, but at $k=6$, the gap statistic decreases. It is this deflection, however brief, that the algorithm is designed to detect, because the goal is to identify the smallest k that is supported by the data. A parsimonious choice of k serves our purposes as well, as we seek to identify a small set of distinct communities with relatively few members to eventually validate in a laboratory setting.

>>COMMENT: 4. The Figure 1B colors would be impossible to differentiate for a person who is colorblind. Some other marker to delineate the differences would be helpful.

RESPONSE: We changed the palette to use colors that would be distinguishably different to people with red/green colorblindness and tested the result using a color blindness simulator.

June 9, 2021

Dr. George O'Toole
Geisel School of Medicine at Dartmouth
Microbiology and Immunology
Rm 202 Remsen
N College St
Hanover, NH 03755

Re: Spectrum00029-21R1 (Mild CF Lung Disease is Associated with Bacterial Community Stability)

Dear Dr. George O'Toole:

Your manuscript has been accepted, and I am forwarding it to the ASM Journals Department for publication. You will be notified when your proofs are ready to be viewed.

Sincerely,

Amanda Oglesby
Editor, Microbiology Spectrum

Journals Department
Supplemental Table 3: Accept

Supplemental Table 1: Accept
Supplemental Table 2: Accept